# Analgosedation in Pediatric Emergency Care: A Comprehensive Scoping Review

**DOI:** 10.3390/ph17111506

**Published:** 2024-11-08

**Authors:** Lorenzo Ciavola, Francesco Sogni, Benedetta Mucci, Eleonora Alfieri, Angela Tinella, Elena Mariotti Zani, Susanna Esposito

**Affiliations:** Pediatric Clinic, Department of Medicine and Surgery, University Hospital of Parma, 43126 Parma, Italy; lorenzociavola@gmail.com (L.C.); francesco.sogni@hotmail.it (F.S.); benedettamucci@gmail.com (B.M.); ele.alfieri95@gmail.com (E.A.); angela.tinella82@gmail.com (A.T.); e.mariottizani@gmail.com (E.M.Z.)

**Keywords:** analgosedation, dexmedetomidine, fentanyl, ketamine, midazolam, nitrous oxide, pain management, pediatric emergency, sedation

## Abstract

Effective management of pain and anxiety in pediatric emergency room is crucial for ensuring both the physical and emotional well-being of young patients. Analgosedation, a combination of analgesia and sedation, is commonly used to facilitate various procedures in children. However, selecting the optimal agent and administration route remains challenging due to the unique pharmacological profiles and side effects of available drugs. This scoping review aims to provide a comprehensive analysis of the pharmacological agents used for procedural analgosedation in pediatric emergency settings, focusing on their efficacy, safety, administration routes, and potential side effects. A systematic review of the literature was conducted, focusing on key agents such as ketamine, midazolam, dexmedetomidine, fentanyl, and nitrous oxide. Studies were included based on their relevance to pediatric procedural sedation, particularly in emergency settings. Literature analysis showed that ketamine and fentanyl are effective for managing moderate to severe pain, with a rapid onset of action. Fentanyl is preferred for acute pain management following fractures and burns, while ketamine and midazolam are commonly used for emergency analgosedation. Dexmedetomidine, which induces sedation similar to natural sleep, is particularly effective in preventing pain and agitation during procedures and is well tolerated in children, especially those with developmental disorders. Nitrous oxide, when used in a 50% oxygen mixture, offers a valuable option for conscious sedation during mildly to moderately painful procedures, maintaining respiratory and airway reflexes. No single drug is ideal for all pediatric patients and procedures and the choice of agent should be tailored to the specific clinical scenario, considering both the sensory and affective components of pain. Future research should prioritize large-scale comparative studies, the exploration of combination therapies, and the development of non-pharmacological adjuncts to enhance the safety and efficacy of pediatric analgosedation.

## 1. Introduction

Analgosedation encompasses a comprehensive approach that combines pharmacological and non-pharmacological strategies to manage pain and anxiety in patients, particularly during medical procedures, without compromising cardiorespiratory function. In pediatric emergency rooms (ERs), managing pain, stress, and anxiety is a crucial aspect of patient care. Acute pain is a frequent reason for pediatric ER visits, and the procedures required during evaluation and treatment can often exacerbate pain or induce anxiety in children.

The unpredictable nature of ER visits, coupled with the unfamiliar environment and potentially distressing interventions, can significantly heighten anxiety and stress in both children and their families. Recognizing this, the effective management of pain and anxiety in pediatric patients is not only considered best medical practice but is also recognized as a fundamental right of the child. Healthcare providers are ethically and professionally obligated to take all necessary measures to prevent pain and minimize its perception, while also addressing the emotional well-being of the child by involving family members in the care process [1,2].

Procedural analgosedation is particularly critical in pediatric emergency settings where common interventions, such as wound suturing, fracture reduction, diagnostic imaging, abscess drainage, and burn management, necessitate a careful balance between efficacy and safety. The choice of analgosedation technique is influenced by the specific characteristics of the procedure, the required depth of sedation, and the patient’s overall health status, including any potential contraindications or risk factors.

This review aims to provide a comprehensive analysis of the primary pharmacological agents used in procedural analgosedation within pediatric emergency settings, examining their efficacy, safety profiles, and routes of administration. A systematic review of the literature was conducted by the Pediatric Clinic at the University Hospital of Parma, Italy, to identify and evaluate existing studies on procedural analgosedation in pediatric emergency care. This review follows a systematic, scoping approach to evaluate the efficacy, safety, and administration of key pharmacological agents used in procedural analgosedation within pediatric emergency settings. The methodology was designed to capture and synthesize current evidence on drugs commonly used in pediatric procedural sedation: ketamine, fentanyl, midazolam, dexmedetomidine, and nitrous oxide. The literature search was conducted using the PubMed database, covering publications from 2003 to December 2023, utilizing terms such as “analgosedation”, “pediatric emergency”, “children”, “ketamine”, “midazolam”, “dexmedetomidine”, “fentanyl”, and “nitrous oxide”. Boolean operators were used to refine and broaden the search scope to capture relevant articles comprehensively. Studies were included if they met the following criteria: (1) focused on pediatric patients aged 0–18 years undergoing procedural analgosedation in emergency settings, (2) investigated the efficacy, safety, or administration routes of the specified drugs, (3) published in peer-reviewed journals, and (4) available in English. Exclusion criteria encompassed studies focusing on adult populations or non-emergency settings, as well as case reports, editorials, or opinion pieces lacking empirical data. Titles and abstracts of all retrieved articles were independently screened by two reviewers (LC and FS) for relevance and eligibility. Discrepancies were resolved through discussion, or by consulting a third reviewer (EA) when necessary. Full-text articles of potentially relevant studies were further assessed against inclusion and exclusion criteria. Data extraction focused on study design, patient demographics, procedure types, drug dosages, administration routes, and outcome measures, including pain relief, anxiety reduction, adverse events, and contraindications. To maintain consistency, standardized outcome metrics (such as pain scores, sedation duration, and recovery times) were prioritized when available. Given the scope of studies and the heterogeneity of designs, a meta-analysis was not feasible. Instead, findings were synthesized narratively to highlight trends in drug efficacy, administration routes, and safety profiles, while identifying gaps for future research. Special attention was given to combination therapies and the comparative effectiveness of drug pairs. Additionally, non-pharmacological adjuncts were noted when integrated with pharmacologic treatments, as these provide potential for enhancing safety and efficacy. Based on the synthesis, recommendations were formulated to guide future research priorities, including the need for longitudinal studies, standardized outcome measures, and optimal dosing strategies for intranasal and other non-invasive administration routes.

## 2. Ketamine

Ketamine is a phencyclidine derivative with a unique profile that makes it highly valuable in pediatric emergency care [3]. Depending on the dose, ketamine is used either for inducing general anesthesia or, at lower doses, as a sedative and analgesic agent. It acts as a noncompetitive antagonist of NMDA glutamate receptors, leading to increased dopamine release in the limbic system due to the inhibition of glutamatergic activation of inhibitory GABAergic pathways. This mechanism can result in disinhibition, causing patients to experience delirium and hallucinations post-anesthesia. Moreover, ketamine reduces dopamine release in the prefrontal cortex, following inhibition of the glutamatergic circuit, leading to decreased memory and cognitive functions, such as amnesia and catatonia. These effects contribute to the state of dissociative anesthesia, where the patient is awake but catatonic and unresponsive to external stimuli.

Ketamine also possesses potent analgesic properties. Beyond its psychotomimetic and amnesic effects, ketamine inhibits catecholamine reuptake, enhancing sympathetic nervous system stimulation, which results in cardiovascular effects such as increased blood pressure, heart rate, cardiac output, and bronchodilation. Importantly, ketamine does not depress respiratory function, making it a preferred anesthetic in various emergency scenarios [4]. Additionally, ketamine increases muscle tone, blood glucose levels, cortisol, and prolactin. A notable adverse effect of ketamine is hypersalivation, which many clinicians mitigate by co-administering a vagolytic agent such as atropine [5].

Ketamine’s lipophilicity and low plasma protein binding facilitate its rapid passage across the blood–brain barrier, leading to brain concentrations that significantly exceed plasma levels. Ketamine is primarily metabolized in the liver to norketamine, which is subsequently hydroxylated and excreted via urine and bile. Although ketamine is primarily administered intravenously, and intramuscularly in some countries when intravenous access is unavailable, it can also be delivered orally, sublingually, and rectally, although these routes have lower bioavailability [6]. Intranasal administration is particularly prevalent in pediatric emergency departments due to its simplicity, non-invasiveness, and good tolerance in children [7].

The intranasal dose of ketamine ranges from 0.5 to 4 mg/kg, producing both analgesic and sedative effects. It is effective in managing moderate to severe pain, with an onset of action within 5–10 min [8]. Unlike opioids, ketamine does not induce histamine release, thus avoiding side effects such as itching and nasal congestion, making it an ideal sedative for asthmatic patients [9].

Yenigum et al. conducted a prospective study that demonstrated the superior efficacy of intranasal ketamine and fentanyl compared to intravenous paracetamol in reducing postoperative pain following tonsillectomy in children [10]. Several studies have compared the analgesic effects of intranasal fentanyl and ketamine for pain relief following pediatric bone fractures. Most conclude that both drugs are effective within 20 min of administration. However, Quin et al. reported that intranasal fentanyl provided faster pain relief than ketamine, showing effects within 10 min [11]. Additionally, ketamine has a slightly higher risk of minor, transient adverse effects, such as a bad taste, dizziness, and drowsiness [12,13].

In 2019, Nemet et al. developed a protocol for analgosedation that includes the use of intranasal fentanyl, ketamine, and midazolam, either alone or in combination, demonstrating their safety and efficacy. Fentanyl is primarily used for acute pain from bone fractures, scalds, and burns, while ketamine and midazolam are favored for emergency analgosedation [14]. Sado-Filho et al. conducted a triple-blind, randomized controlled trial in 2019, showing that a combination of intranasal ketamine and midazolam was more effective than midazolam alone in managing uncooperative children undergoing dental procedures [15].

Intranasal midazolam, ketamine, and oral chloral hydrate have also been effective for sedating children undergoing echocardiography, with ketamine providing shorter sedation durations [16]. Retrospective studies indicate that the combination of intranasal ketamine and dexmedetomidine during diagnostic procedures (e.g., echocardiography, EEG, MRI) is effective, with a low incidence of adverse effects [4]. Moreover, adding ketamine can reduce negative cardiovascular effects, such as bradycardia and hypotension, associated with dexmedetomidine [17,18]. Nielsen et al. reported that intranasal ketamine combined with sufentanil in children undergoing painful procedures like chest tube removal and burn debridement induces rapid analgesia without causing desaturation or significant changes in heart rate [19].

## 3. Midazolam

Midazolam is a γ-aminobutyric acid (GABA) receptor agonist that exerts sedative, anxiolytic, and amnesic effects [20]. It has been extensively used as a premedication in pediatrics through various routes, including oral, nasal, rectal, intravenous (IV), and intramuscular (IM). However, the IV route requires venous access, and the IM route can be painful, making oral and intranasal administration preferable in children due to their safety, effectiveness, and ability to provide adequate sedation and calm separation from parents [21].

A study by Manoj et al. compared the administration ease of oral versus intranasal midazolam when given by parents to children undergoing elective surgery. The results showed that oral midazolam was easier for parents to administer and more acceptable to children than the intranasal form [21]. Oral midazolam, at a dose of 0.5 mg/kg, is commonly used as the premedication of choice in young children before dental procedures [22]. In older, anxious, or combative children, intranasal dexmedetomidine provides satisfactory sedation and smooth separation from parents and mask induction, owing to its safety, ease of administration, and minimal impact on hemodynamics and respiration [22].

The intranasal dose of midazolam ranges from 0.2 to 0.5 mg/kg [23]. Therapeutic levels in the cerebrospinal fluid are reached more rapidly with intranasal administration due to the rich vascular plexus of the nasal cavity, which communicates with the subarachnoid space via the olfactory nerve [24]. This route obviates the need for IV access and is readily accessible [24]. Brown et al. reported that in children with autism spectrum disorders in the ED, intranasal midazolam is among the most commonly used sedatives, alongside IV ketamine [25].

Alp et al. conducted a study comparing the sedative effects of intranasal ketamine, intranasal midazolam, and oral chloral hydrate in children undergoing transthoracic echocardiography. The sedation was considered successful if it resulted in high-quality imaging. The study found that all three agents achieved successful sedation, with intranasal midazolam having the fastest onset and intranasal ketamine providing the shortest duration of sedation but with fewer side effects [16].

The adverse effects of intranasal midazolam include nasal irritation, a bitter taste, and vomiting [23]. Chiaretti et al. evaluated the safety and efficacy of administering intranasal lidocaine before intranasal midazolam in anxious, uncooperative children undergoing minor painful or diagnostic procedures. The combination was found to prevent nasal discomfort and provide satisfactory sedation, meeting the expectations of both parents and medical staff. Although further studies are required, this approach may be effective for sedating children undergoing minor procedures or diagnostic investigations [24]. Additionally, combining ketamine with midazolam appears more effective than midazolam alone, as demonstrated in a study comparing intranasal versus oral ketamine and midazolam during dental treatments [15]. Furthermore, the combination of intranasal midazolam and fentanyl has proven effective for procedural sedation in young children undergoing simple laceration repair [26,27].

Midazolam can cause respiratory and circulatory depression, though these side effects are infrequent when the drug is used alone [23]. Notably, midazolam is considered safe even with short fasting times. A study by Malia et al. showed that even children with “nothing by mouth” times of less than 2 h for solids and liquids experienced no adverse outcomes [28]. This represents a significant advantage in emergency settings, where intranasal sedatives require less staff involvement, less physical restraint of children, and faster sedation achievement [28].

Table 1 outlines the dosage and the characteristics of action of midazolam according to the route of administration.

## 4. Dexmedetomidine

Dexmedetomidine is an α2-adrenoceptor agonist primarily utilized for its sedative, anxiolytic, and analgesic properties. As the S-enantiomer of medetomidine, it binds to both central and peripheral α2-receptors to exert its effects [29]. Dexmedetomidine can be administered through various routes, including intranasal, oral, intramuscular, and intravenous, with dosages ranging from 0.5 µg/kg to 4 µg/kg depending on the level of sedation required for the procedure [23]. Intranasal administration is particularly favored due to its rapid absorption compared to oral delivery [30,31]. However, the onset of action remains relatively slow, typically taking 15–30 min, with a half-life of approximately 2 h. The sedation induced by dexmedetomidine closely resembles a natural sleep state [32,33].

Intranasal dexmedetomidine has shown to be well tolerated in children and, in some cases, even more effective than intranasal or oral midazolam for preventing pain and agitation during procedures like intravenous insertion and dental treatments [34]. The drug is particularly suitable for pediatric sedoanalgesia as it is odorless and tasteless, making it more acceptable to infants and older children. Another advantage is that, even at higher doses, patients remain easily arousable with simple external stimuli. Moreover, dexmedetomidine preserves protective airway reflexes and respiratory drive, maintaining airway patency, which distinguishes it from other sedative agents [31,35]. However, it is important to note that adverse effects, such as bradycardia, hypotension, and occasionally hypertension, may occur depending on the dose administered [35].

Dexmedetomidine has been successfully used in a variety of clinical and surgical settings for pediatric patients, including diagnostic radiologic procedures like computed tomography (CT) and magnetic resonance imaging (MRI), as well as more invasive procedures such as central venous catheter placement, bronchoscopy, laryngoscopy, cardiac catheterization, and awake craniotomies [36,37].

Dexmedetomidine has also shown promise in managing sedation in children with autism spectrum disorder (ASD). In a case series by Carlone et al., intramuscular dexmedetomidine (initial dose of 4 µg/kg) was administered to eight children with ASD (ages 5–14 years) requiring urgent diagnostic procedures in the emergency department. Sedation was successfully achieved in seven of the eight patients with the initial dose; only one patient required a supplemental dose of 1 µg/kg. The mean induction time was 30 min, with a mean recovery time of 135 min. Notably, no adverse effects such as bradycardia, hypotension, or respiratory complications were observed, though the authors emphasized avoiding painful intramuscular injections when less invasive options are available [38].

Several studies have compared dexmedetomidine with other commonly used sedatives in the emergency room. For example, Azizkhani et al. evaluated the combination of intravenous dexmedetomidine and ketamine versus propofol and ketamine in reducing recovery agitation in children undergoing sedation for painful procedures. The study found that the dexmedetomidine and ketamine combination significantly reduced recovery agitation, including symptoms such as hallucinations, crying, and nightmares, compared to the other groups. Although transient hypotension was more common in the dexmedetomidine group, there were no significant differences in peripheral oxygen desaturation among the groups. The authors concluded that dexmedetomidine combined with ketamine could be an effective option for procedural sedation in children [39].

Another study compared intranasal dexmedetomidine (3 µg/kg) with intranasal midazolam (0.3 mg/kg) in 162 children aged 1 to 6 years requiring sedation for CT imaging. While dexmedetomidine resulted in greater decreases in heart rate and mean blood pressure, it provided faster and more satisfactory sedation according to clinicians. However, due to its cardiovascular effects, dexmedetomidine may not be suitable for children with hemodynamic instability [40].

Neville et al. conducted a double-blind randomized controlled trial in children aged 1 to 5 years needing suture repairs in the emergency room. They found that while both intranasal dexmedetomidine (2 µg/kg) and midazolam (0.4 mg/kg) were effective for sedation, dexmedetomidine was superior in reducing anxiety during positioning for the procedure. No serious adverse effects were observed in either group [41].

A trial conducted in two trauma centers in Iran compared dexmedetomidine and ketamine for sedation in children aged 2–14 years undergoing laceration repair. The study highlighted that dexmedetomidine is useful when a shorter sedation time is needed, as its effects are faster, while ketamine provided a shorter recovery time [42].

Oriby studied the effects of different sedatives, including intranasal midazolam, oral ketamine, and intranasal dexmedetomidine, in children aged 2–6 years undergoing dental procedures. The combination of dexmedetomidine and ketamine resulted in faster sedation onset and more satisfactory post-procedure analgesia, while midazolam was associated with more side effects, such as shivering [20].

Table 2 outlines the dosage and characteristics of action of dexmedetomidine according to the route of administration.

## 5. Fentanyl

Fentanyl is a potent synthetic opioid known for its rapid onset of action due to its high lipophilicity and low molecular weight. It is commonly used in pediatric emergency settings for pain management, including procedures such as sutures, fracture reduction, burn debridement, foreign body removal, and abscess incision. Fentanyl is often preferred over morphine in children due to its rapid action and comparable efficacy with minimal sedation and hemodynamic impact [43]. Despite the widespread use of fentanyl and its derivatives (such as sufentanil, alfentanil, and remifentanil), pharmacokinetic data in pediatric populations are limited. Intranasal fentanyl (INF) is frequently used in pediatric emergencies at a standard dose of 1–2 µg/kg for premedication, emergency analgesia, and palliative care [44]. Table 3 compares the characteristics of fentanyl and its derivatives.

Intravenous fentanyl and sufentanil are typically used intraoperatively during general anesthesia, while alfentanil and remifentanil are favored for short or outpatient procedures requiring analgo-sedation. Fentanyl is increasingly replacing morphine due to its convenience and comparable safety profile. Studies have shown no significant difference in pain scores between INF and intravenous morphine for limb fractures and burns, although INF may provide faster analgesia [45]. For instance, Holdgate et al. found that INF had an onset time of 32 min compared to 63 min for intravenous morphine (*p* = 0.001) [46]. Another study demonstrated that INF provided effective analgesia within 10 min of administration in pediatric patients with painful orthopedic trauma [47]. INF has also proven effective for postoperative pain and agitation relief [48].

No serious adverse effects have been reported following INF use [49]. A prospective study on the effectiveness of INF in children aged 1–3 years at a dose of 1.5 µg/kg showed a significant decrease in FLACC (face, legs, activity, cry, consolability) scores with no adverse drug reactions [50]. However, some studies have noted a “bad taste” and an increased incidence of vomiting with INF compared to placebo (3.9–12%) [51].

Although studies comparing premedication with INF before N_2_O 70% inhalation and placebo have not shown significant differences in pain scores, there was an increased incidence of vomiting and deeper sedation levels when INF was used in conjunction with N_2_O 70% [52]. The NICE guidelines suggest that combining INF with midazolam may be a safe and effective strategy for procedural sedation in pediatric patients undergoing minor procedures, such as laceration repair or orthopedic manipulation, especially in urgent care settings [27,53,54].

## 6. Nitrous Oxide

Nitrous oxide is an anesthetic gas naturally present in the atmosphere, known for its sedative, anxiolytic, moderately analgesic, and amnesic properties. Although its precise mechanism of action remains unclear, it is believed that its analgesic effects are mediated through modulation of opioid receptors. The effects of nitrous oxide are dose dependent, impacting cognitive and sensory functions at concentrations as low as 15%. At higher concentrations, exceeding 60–70%, it can induce loss of consciousness.

When inhaled, nitrous oxide is rapidly absorbed through the alveoli and is almost entirely eliminated via exhalation, making it particularly suitable for patients with hepatic or renal insufficiency. Additionally, it does not undergo biotransformation in tissues, reducing the risk of pharmacological interactions [55,56]. The effects of nitrous oxide typically begin within 3 min of inhalation and subside within 5 min of discontinuation.

Inhalation of a 50% nitrous oxide/50% oxygen mixture induces a state of conscious sedation, where the patient experiences a depressed level of consciousness while maintaining independent breathing, protective airway reflexes, and the ability to respond to verbal stimuli. This allows for its use without the need for fasting or intravenous access [57]. Outside the operating room, nitrous oxide is commonly used in a 50–50 mixture, providing sufficient analgesic and anxiolytic effects for moderately painful procedures. This dosage has been shown to be highly effective and safe in numerous pediatric studies, making it a preferred agent for short procedures that involve mild to moderate pain. Notably, it can be safely administered by adequately trained nursing staff [58].

Administration of nitrous oxide can be continuous or on-demand using a facial mask, with the latter activated by the patient through deep inspiration, making it suitable only for cooperative children [59,60]. Side effects are rare and typically resolve quickly once administration is stopped. Minor side effects, reported in approximately 5% of patients, include disinhibition, disorientation, dizziness, headache, euphoria, restlessness, nausea, and vomiting. Major side effects are extremely rare (0.3%) and are more likely to occur when nitrous oxide is used in concentrations above 50% or in combination with benzodiazepines or opioids. These more serious side effects include desaturation, apnea, airway obstruction, bradycardia, and loss of verbal contact [61,62].

The contraindications for nitrous oxide use are relatively few but are important to consider, especially in conditions where inhaled gas can rapidly diffuse into air-filled spaces, leading to increased pressure. Such conditions include pneumothorax, pulmonary emphysema, pulmonary hypertension, respiratory distress, closed head trauma, intracranial hypertension, otitis, sinusitis, and recent middle ear or retinal surgery [55,63].

The efficacy of nitrous oxide is well supported in the literature. A prospective, observational, multicenter study by Gomez et al. included 213 patients aged 2 to 18 years who received nitrous oxide for painful procedures in pediatric emergency settings [64]. The most common procedures included wound repair (24.4%), fracture or dislocation reduction (19.7%), and lumbar punctures (18.3%). The study found that patient behavior was rated as “good/very good” in 79.7% of cases, with no significant differences based on the type of procedure performed. Adverse effects were recorded in 7.9% of cases, most commonly dizziness and headache. The medical team found the administration process easy in 96.6% of cases, and 92.7% of parents indicated they would accept the use of nitrous oxide in similar future situations for their child.

## 7. Conclusions

While no single drug possesses all the ideal characteristics for analgesia and sedation, optimizing the choice of an analgesic requires a comprehensive understanding of both the sensory and affective components of pain. Tailoring the analgesic approach to the specific causes and types of pain is crucial in achieving the best outcomes for pediatric patients.

Intranasal administration of analgesics is gaining popularity due to its efficacy, ease of use, and growing support from medical evidence. The side effects associated with this route are generally mild and comparable to those observed with intravenous administration.

Ketamine and fentanyl have proven to be effective analgesics for managing moderate to severe pain, with a rapid onset of action within 5–10 min. Fentanyl is particularly effective for treating acute pain from bone fractures and burns, while ketamine and midazolam are commonly used for emergency analgosedation. Midazolam is well tolerated in its oral form by both children and parents, making it a preferred premedication for dental procedures and simple laceration repairs. Although midazolam is generally safe, especially when used alone, it can still pose risks of respiratory and circulatory depression.

Dexmedetomidine offers sedation that closely resembles natural sleep and is more effective than midazolam in preventing pain and agitation during procedures such as intravenous insertion and dental treatments. Its odorless and tasteless properties make it particularly suitable for difficult cases, such as children with autism, and for urgent radiodiagnostic procedures. However, it carries a higher risk of bradycardia and hypotension compared to midazolam.

The equimolar nitrous oxide/oxygen mixture provides effective analgesia and anxiolysis, making it a valuable option for cooperative children experiencing mild to moderate pain during procedures like wound repair, fracture or dislocation reduction, and lumbar punctures. This mixture induces a state of conscious sedation while preserving spontaneous breathing and airway reflexes, eliminating the need for fasting or intravenous access. Although side effects are rare and typically resolve quickly after discontinuation, more severe complications can occur when nitrous oxide is combined with midazolam or fentanyl, making such combinations less advisable.

Despite the effectiveness of these pharmacological options, it remains essential to explore complementary strategies to enhance procedural analgosedation in the emergency room. Healthcare providers involved in pediatric procedural sedation should be well versed in age-appropriate psychological approaches, which are crucial for addressing the emotional needs of both the child and their family members during these procedures. This holistic approach ensures that both the physical and emotional aspects of pain and anxiety are effectively managed, leading to better overall patient care.

Unfortunately, the available literature has some limitations. Expanding the comparative studies of drug combinations (e.g., ketamine with midazolam or fentanyl) versus single agents would offer clearer insights into optimal treatment protocols. Additionally, integrating data on non-pharmacological interventions alongside pharmacologic treatments could provide a more holistic approach to pediatric sedation. Lastly, promoting standardized outcome metrics across studies would aid in achieving more reliable, comparable findings.

In addition, the environment of a pediatric ER can be particularly overwhelming and anxiety inducing for children due to unfamiliar sounds, faces, and procedures. In these scenarios, the presence of parents or familiar caregivers can have a powerful, dual-sided effect. On one hand, a familiar face often provides comfort, helping the child feel safer and more reassured. Studies have shown that calm and supportive caregivers can significantly reduce a child’s anxiety, which in turn can lower the sedation levels required for procedural analgosedation [65]. However, caregivers displaying visible anxiety or distress can inadvertently increase the child’s own anxiety, as children frequently mirror the emotions of those they trust most. This anxiety may lead to increased sedative intervention, as the child becomes more resistant or fearful [66]. The presence of specialized pediatric nurses in the ER is equally critical. In countries where pediatric nursing specialization is prioritized, the ER staff includes nurses trained not only in clinical skills but also in child-specific communication and psychological techniques that help de-escalate a child’s anxiety. These nurses appropriate language, distraction techniques, and emotional reassurance to help children cope more effectively during medical procedures, which can lead to a smoother, less traumatic experience and potentially reduce the overall sedative doses required. Studies have demonstrated that specialized pediatric nurses can positively affect children’s coping strategies and outcomes in emergency settings [67]. In countries without specialized pediatric nurses, children are more likely to experience a generalized care approach, where staff may lack the training needed to address their unique psychological and emotional needs. The absence of such tailored care can lead to increased stress and procedural difficulties, potentially necessitating higher sedative doses to achieve comfort and compliance. Therefore, the presence of specialized pediatric nurses with appropriately managed parental involvement is likely essential in determining sedation type and dosage [68].

In the last years, a lack of research interest in pediatric pain management has been observed. This is troubling, as it may lead to gaps in both expertise and advancements in pediatric care, potentially compromising the development of tailored pain management strategies and innovations in pediatric procedural sedation essential for the well-being of young patients. Future research in pediatric analgosedation should focus on several key areas to enhance the safety, efficacy, and overall patient experience. First, there is a need for large-scale, multicenter trials comparing different analgesic and sedative agents, particularly in terms of their long-term safety profiles, efficacy across various age groups, and impact on different types of procedures. These studies should also explore optimal dosing strategies for intranasal and other non-invasive routes of administration to minimize side effects while maximizing therapeutic benefits. Second, further investigation into the use of combination therapies is essential, particularly in understanding the interactions between drugs like ketamine, midazolam, dexmedetomidine, and nitrous oxide. These studies should aim to establish clear guidelines on which combinations offer the best balance of efficacy and safety for specific procedures. Another critical area for future research is the development and validation of non-pharmacological adjuncts to analgosedation, such as psychological interventions, distraction techniques, and the use of virtual reality. These approaches have the potential to reduce the need for higher doses of sedative agents and minimize adverse effects, particularly in vulnerable populations such as children with chronic conditions or developmental disorders. Moreover, future research should evaluate the impact of caregiver presence and pediatric nursing expertise on sedation outcomes. Finally, more studies are needed to assess the long-term psychological and developmental impacts of repeated exposure to sedation in pediatric patients. Understanding these effects will be crucial in guiding clinical practice, particularly in determining the frequency and circumstances under which sedation is administered. By addressing these priorities, future research can contribute significantly to improving the standards of care in pediatric analgosedation, ensuring safer, more effective, and more patient-centered practices in emergency and procedural settings.

## Figures and Tables

**Table 1 pharmaceuticals-17-01506-t001:** Dosage and characteristics of action of midazolam according to the route of administration.

Route of Administration	Recommended Dosage	Onset of Action	Duration
Oral	0.25–0.5 mg/kg (max 20 mg)	10–20 min	Variable
Intranasal	0.2–0.3 mg/kg (maximum, 10 mg) (use only in patients >6 months of age)	5 min	30–60 min
Intravenous	0.05–0.1 mg/kg (maximum, 10 mg)	1–5 min	20–30 min

**Table 2 pharmaceuticals-17-01506-t002:** Dosage and characteristics of action of dexmedetomidine according to the route of administration.

Route	Usual Dose Range	Onset of Action	Time to Peak Effect	Notes
IV bolus and infusion	1 µg/kg bolus–0.2–1.5 µg/kg infusion	5–10 min	15–30 min	Bolus dosing may be associated with increased risk of hypotension and bradycardia
IV infusion alone	0.2–1.5 µg/kg/h infusion	15 min	60 min	Doses above 1.5 µg/kg/h demonstrate no additional sedative effect
Intranasal	1–4 µg/kg	10 min	20 min	
Intramuscular	1–4 µg/kg	15–20 min	Unclear	Pharmacodynamics not well studied
Sublingual	120 or 180 µg	45–60 min	60–120 min	Pharmacodynamics not well studied

**Table 3 pharmaceuticals-17-01506-t003:** Characteristics of fentanyl and its derivatives.

	Fentanyl	Sufentanil	Alfentanil	Remifentanil
Potency compared to morphine	100–300	800–1000	40–50	100–200
IV induction dose (μg/kg)	2–6	0.25–2.0	25–100	1–2
IV maintenance dose (μg/kg)	0.5–2	2.5–10	5–10	0.1–1.0
IV infusion rate (μg/kg/h)	0.5–5	0.5–1.5	30–120	0.1–1.0
Other routes of administration than IV	transdermal, transmucosal (buccal, nasal, sublingual), epidural	epidural, sublingual		
Time to onset (min)	1.5	1	0.75	<1
Time to peak effect (min)	4.5–8	2.5–5	1.5	1.5
Duration of peak effect (min)	20–30	30	15	
Duration of analgesic effect (min)	60–120	100–150	30–60	5–10
Analgesic plasma concentration (ng/mL)	0.6–3.0	0.5–2.5	50–300	0.3–3
Plasma concentration associated with loss of consciousness (ng/mL)	>20.0	>2.5	>400	>4
t1/2α (min)	1.7 ± 0.1	1.4 ± 0.3	1.31 ± 0.48	1
t1/2β (min)	13.4 ± 1.6	17.7 ± 2.6	9.4 ± 2.7	6
t1/2γ (min)	219 ± 10 (120–240)	164 ± 22 (120–180)	93.7 ± 8.3 (60–120)	10–20 (6–14)
Vdc (L/kg)	0.36 ± 0.07	0.16 ± 0.02	0.12 ± 0.04	0.1
Vdss (L/kg)	4.0 ± 0.4 (3–5)	1.7 ± 0.2 (2.5–3.0)	1.0 ± 0.3 (0.4–1.0)	0.35 (0.2–0.4)
CL (mL/min/kg)	13 ± 2 (10–20)	12.7 ± 0.8 (10–15)	7.6 ± 2.4 (3–9)	40 (30–60)
Protein binding (%)	80–84	91–92.5	88.7–92.1	70
pKa	8.4	8.0	6.5	7.1
Metabolism	CYP3A	CYP3A	CYP3A	Plasma and tissue esterases
Non-ionized fraction @ pH 7.40 (%)	8.5	20	89	67
Lipid solubility (octanol/water distribution coefficient)	813–816	1727–1778	128	18

Abbreviations: t1/2α distribution half-life, t1/2β redistribution half-life, t1/2γ terminal elimination half-life, Vdc volume of distribution of the central compartment, Vdss volume of distribution at steady state, CL clearance.

## Data Availability

Not applicable.

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
