# Peer review of "Analgosedation in Pediatric Emergency Care: A Comprehensive Scoping Review"

_pharmaceuticals, 2024, doi:10.3390/ph17111506_

Round 1
Reviewer 1 Report
Comments and Suggestions for Authors
Thank you for the opportunity to review this interesting paper. It is well written and easy to read,
This is a narrative review which collects data from a number of studies and presents summaries. It would be a useful starting point for anyone interested in the subject, but it is as it claims, only a narrative review. It does not comment on the design or methodology of the studies or combine studies to strengthen findings.
Having worked in paediatrics in three countries and with close links to three others, two important issues are mentioned but could be developed further; the presence of parents/familiar carers at the time of the procedure, and whether the jurisdiction in which the study was conducted has specialist paediatric nurses, and whether they are present in the emergency room.
I am sure the authors are aware that In what is an unfamiliar and frightening environment for children, the presence of familiar faces can make a huge difference (positive or negative if for example, the parents are anxious), but the presence of nursing staff who are trained and experienced with children can make a huge difference to how the child responds to the situation, and how traumatic or otherwise the experience.
The presence or absence of such nursing expertise probably makes a significant contribution to the sedation required.
I am somewhat surprised by the comment in line 110 that intramuscular administration is common. In all paediatric environments of my experience, the use of the intramuscular route would be regarded as very poor practice unless there was no other alternative.
I think it would be useful to comment upon these points within the discussion. Otherwise I would welcome publication.
Comments on the Quality of English LanguageGenerally very high standard only minor editing required.
Author Response
Thank you for the opportunity to review this interesting paper. It is well written and easy to read.
Re: Thank you for your positive evaluation. We improved the manuscript according to your suggestions.
This is a narrative review which collects data from a number of studies and presents summaries. It would be a useful starting point for anyone interested in the subject, but it is as it claims, only a narrative review. It does not comment on the design or methodology of the studies or combine studies to strengthen findings.
Re: We improved the Methods section according to your comments (pp. 2-3) and we added a comment at this regard in the conclusive sections (p. 13).
Having worked in paediatrics in three countries and with close links to three others, two important issues are mentioned but could be developed further; the presence of parents/familiar carers at the time of the procedure, and whether the jurisdiction in which the study was conducted has specialist paediatric nurses, and whether they are present in the emergency room.
I am sure the authors are aware that In what is an unfamiliar and frightening environment for children, the presence of familiar faces can make a huge difference (positive or negative if for example, the parents are anxious), but the presence of nursing staff who are trained and experienced with children can make a huge difference to how the child responds to the situation, and how traumatic or otherwise the experience.
The presence or absence of such nursing expertise probably makes a significant contribution to the sedation required.
Re: We included a section on the role of parents and specialized pediatric nurses in the Conclusions (p. 13) with four new references (p. 19).
I am somewhat surprised by the comment in line 110 that intramuscular administration is common. In all paediatric environments of my experience, the use of the intramuscular route would be regarded as very poor practice unless there was no other alternative.
Re: The sentence has been revised according to your comment (p. 3).
I think it would be useful to comment upon these points within the discussion. Otherwise I would welcome publication.
Re: We revised the text according to your comments and we hope that you could consider our revised manuscript suitable for publication.
Reviewer 2 Report
Comments and Suggestions for Authors
The manuscript includes carefully selected literature according to clear criteria. The review indicates a low interest of researchers in pain management among children. This is a very worrying phenomenon, because pediatrics and other medical specialties such as pediatric cardiology, pediatric surgery and even more so anesthesiology are not popular fields among young doctors.
Comment:
The manuscript submitted for review addresses the topic of the use of painkillers in emergency interventions in children. This is a particularly important problem in the case of the need for surgical interventions, treatment of injuries and imaging diagnostics in young patients. Pediatric practice requires obtaining effective analgesia with a certain sedative component, which will allow the patient to calm down in emergency situations accompanied by fear and pain. The authors reviewed the latest studies, the selection of literature for the review was determined very factually and precisely, which deserves special recognition. Collecting information on interventional analgesia in the review is justified and allows practitioners to make an informed choice of the method of administration and type of drugs.
I recommend publication without significant changes, because the review was conducted reliably, described in a logical manner without significant errors. I do not see the need for changes in the methodology or the way of presenting the substantive content. Ketamine, fentanyl and other anesthetics were compared in terms of effectiveness, safety and dose. While the use of ketamine itself, which has not been the standard choice for induction of general anesthesia in humans in recent decades due to the side effects associated with dissociative anesthesia, has not been met with enthusiasm, there are growing reports of its use on mucous membranes as an analgesic that is easy to administer in emergency medical interventions in emergency departments, also for analgesia in the case of myocardial infarction or circulatory collapse. Low doses of ketamine do not impair respiratory or circulatory functions. Local administration of ketamine in children in emergency situations does not generate the additional anxiety associated with injection and allows for drug distribution even in the case of vomiting. The reports included in the manuscript have been properly discussed, summarized and provided with an accurate conclusion regarding the need to search for an appropriately combined multi-drug therapy, because none of the available drugs is an ideal drug, but used in an appropriate combined therapy in safe doses may be helpful in obtaining the desired effect: analgesic and anxiolytic effects, enabling imaging diagnostics and necessary surgical intervention.
Author Response
The manuscript includes carefully selected literature according to clear criteria. The review indicates a low interest of researchers in pain management among children. This is a very worrying phenomenon, because pediatrics and other medical specialties such as pediatric cardiology, pediatric surgery and even more so anesthesiology are not popular fields among young doctors.
Re: Thank you for your positive evaluation. We revised the text highlighting your comment (p. 13) and revising it also according to the other reviewer’s suggestions.